# Testosterone Therapy for Late-Onset Hypogonadism: A Clinical, Biological, and Analytical Approach Using Compounded Testosterone 0.5–20% Topical Gels

**DOI:** 10.3390/pharmaceutics16050621

**Published:** 2024-05-06

**Authors:** Daniel Banov, Bruce Biundo, Kendice Ip, Ashley Shan, Fabiana Banov, Guiyun Song, Maria Carvalho

**Affiliations:** Professional Compounding Centers of America (PCCA), Houston, TX 77099, USA

**Keywords:** hormone supplement (replacement) therapy, testosterone, late-onset hypogonadism, androgen deficiency, case study, skin permeation, physicochemical stability, personalized preparations, pharmacy compounding

## Abstract

Testosterone is integral to men’s sexual and overall health, but there is a gradual decline in the ageing male. The topical administration of testosterone is a valuable option as a supplement (replacement) therapy to alleviate hypogonadal symptoms. The clinical efficacy of a compounded testosterone 5% topical gel was assessed retrospectively in a male patient in his seventies by evaluating the laboratory testing of the serum total testosterone and the results of a validated androgen deficiency questionnaire. After treatment, the patient’s hypogonadal symptoms improved and the serum total testosterone level achieved was considered clinically optimal. The skin permeation of the testosterone topical gel (biological testing) was evaluated in vitro using the Franz finite dose model and human cadaver skin, and it is shown that testosterone can penetrate into and through ex vivo human skin. Testosterone therapy is often prescribed for extended periods, and consequently, it is crucial to determine the beyond-use date of the compounded formulations. The analytical testing involved a valid, stability-indicating assay method for compounded testosterone 0.5% and 20% topical gels. This multidisciplinary study shows evidence supporting topically applied testosterone’s clinical efficacy and the compounded formulations’ extended stability. Personalized, topical testosterone therapy is a promising alternative in current therapeutics for hypogonadal patients.

## 1. Introduction

Male hypogonadism is a broad term that refers to androgen deficiency or low testosterone in men. Andropause is a common form of hypogonadism that refers to the gradual testosterone decline in the ageing male. This decline is predominantly due to a failure in the function of the hypothalamic–pituitary unit, and thus, it is also referred to as late-onset hypogonadism [1]. The prevalence of andropause is difficult to estimate due to the heterogeneity of the populations and methodologies used in the studies, as well as a lack of consistency of reference biochemical values [2]. However, it is suggested that it is a widespread condition that currently affects over one-quarter of men in the United States. Studies indicate that the prevalence of andropause is likely to increase due to the ageing population and co-morbidities such as obesity, diabetes, dyslipidemia, hypertension, metabolic syndrome, and chronic obstructive pulmonary disease (COPD) [3,4,5].

Testosterone plays an integral role in men’s sexual and overall health. As such, androgen deficiency is linked to a complexity of symptoms, such as erectile dysfunction, loss of libido, depressed mood, lethargy, osteoporosis, and declining muscle tone. It is essential to consider testosterone supplement (replacement) therapy to improve the quality of life and long-term health of late-onset hypogonadal patients. There are several options for testosterone supplementation, such as implantable pellets, parenteral dosing, sublingual or buccal administration, and topical application. Orally administered testosterone suffers an extensive first-pass effect and it is associated with liver toxicity. Pellets are implanted at the doctor’s office; it is an invasive procedure that lacks the flexibility of dosing adjustments. Parenteral dosing via intramuscular (IM) injection is a common form of administering testosterone, but there are difficulties in achieving a steady-state release upon dosing. Subcutaneous administration is preferable, better tolerated, and more convenient than IM. Sublingual or buccal administration, although non-invasive and effective, requires multiple dosing throughout the day, which often leads to poor compliance. Currently, topical application appears to be the most physiologic route of dosing as it allows for a continual, steady-state release of testosterone. Topical gels, in particular, at concentrations of testosterone from 1% to 5% have proven to be effective and user-friendly. Topical application is a simple procedure and attention must only be paid to the potential transfer of testosterone to a child or female. Furthermore, compounded topical gels may be customized throughout the treatment to the meet the patient’s variable dosing needs [6,7].

This multidisciplinary study aims to test compounded testosterone gels’ clinical efficacy, skin permeation, and physicochemical stability. The clinical efficacy is tested in a patient case study, whereas the skin permeation is tested in vitro in a biological laboratory. The physicochemical stability testing is performed in an analytical laboratory. The purpose of this study is to generate clinical and scientific evidence to support testosterone topical therapy in hypogonadal patients.

The case study concerns a male in his seventies with persistent late-onset hypogonadal symptoms. A physician prescribed compounded testosterone 5% topical gel in Atrevis Hydrogel^®^ (PCCA, Houston, TX, USA), a proprietary topical base designed to deliver testosterone through the skin in male patients [8]. The compounded testosterone gel is dispensed in a 75 mL MegaPump^®^ (PCCA), which delivers approximately 0.5 mL of gel per pump. The formula, method of preparation, and storage instructions for the topical gel are displayed in Table 1. The patient applied two pumps of the gel every morning on the inner forearm, thus approximately 50 mg of testosterone daily for six months, as directed by his physician. The patient provided a written consent for the publication of the results. The purpose of this case study is to discuss the management of hypogonadal symptoms with topical testosterone.

The biological testing consisted of evaluating the permeation of a compounded testosterone 10% topical gel over 30 h from a single application on human cadaver skin. The in vitro permeation test (IVPT) was the model used applying the finite dose technique and Franz diffusion cells to dose and culture the skin samples. Data defining the total absorption, absorption rate, and skin content can be accurately determined using the IVPT. This model is a valuable tool in predicting the in vivo percutaneous absorption kinetics of topically applied drugs [9,10].

Testosterone therapy is often prescribed for extended periods, and consequently, it is essential to determine the beyond-use date (BUD) of the compounded topical gels. The analytical testing consisted of a valid, stability-indicating assay method developed for compounded testosterone 0.5% and 20% topical gels. The physicochemical stability was tested for six months.

## 2. Materials and Methods

### 2.1. Case Study

The clinical efficacy of the compounded testosterone gel was assessed retrospectively by inviting the patient to complete the Androgen Deficiency in the Ageing Male (ADAM) questionnaire, a non-invasive screening test to detect low testosterone in males over 40 years of age. It is a self-reported questionnaire with ten ‘Yes/No’ questions, from libido to work performance. The questionnaire was validated in the general population and was found to have 88% sensitivity and 60% specificity for detecting androgen deficiency. According to Morley et al. [1], the patient is likely to have low testosterone if he answers ‘Yes’ to question numbers 1 or 7 or if he answers ‘Yes’ to more than three questions. This primary outcome measure aims to report the change from baseline in the self-reported symptoms of androgen deficiency. As such, the patient was invited to complete the ADAM questionnaire twice by referring to his late-onset hypogonadal symptoms before treatment (baseline) and after treatment.

The secondary outcome measure of this case study was the laboratory testing of the patient’s serum total testosterone by the end of the six months of compounded testosterone therapy. The patient was tested in the fasting state on a morning sample. The laboratory testing aims to evaluate the patient’s hormone levels, which, together with the clinical assessment, aid in diagnosing and managing the condition. The serum total testosterone is the standard evaluation of androgen status, routinely used in clinical practice [11].

### 2.2. Biological Testing

The permeation of testosterone was evaluated in vitro by Diteba (Toronto, ON, Canada) according to an internal protocol (DTM-179-R00), which details the method of analysis, skin preparation, dose administration, and sample collection of the study. The IVPT model consists of human torso skin mounted on modified Franz diffusion chambers that maintain the skin at a temperature and humidity that match typical in vivo conditions. A finite formulation dose is applied to the skin’s outer surface, and drug absorption is measured by monitoring its appearance rate in the receptor solution bathing the inner surface of the skin (transdermal flux) [9,10]. The drug distribution is measured by analyzing the skin content of the samples by ultra-performance liquid chromatography coupled with ultraviolet detection (UPLC-UV).

The dermatomed cryopreserved skin samples were obtained from three donors (Science Care, Phoenix, AZ, USA), all serologically tested and free of infectious diseases. The frozen skin was thawed at room temperature, cut into small sections, and soaked in a diffusion medium for at least 30 min. The samples were mounted on modified Franz cells with the stratum corneum facing upward. All diffusion cells were mounted in a diffusion apparatus and the diffusion medium (phosphate-buffered saline containing 0.1% Tween 20 and 0.008% gentamicin sulfate) was added to the receptor compartment. Before dose administration, the skin integrity test (transcutaneous electrical resistance) was performed using the Precision LCR meter and the tissues with electrical resistance 3 times higher than the reading of the diffusion medium were used.

Ten formulations of testosterone 10% in Atrevis hydrogel (PCCA, lot 20170518) were accurately weighed (8.8 mg ± 20%) and applied to a total of nine skin sections (three replicates per donor). The diffusion medium in the receptor compartment was stirred magnetically at approximately ~600 revolutions per minute (rpm) and the temperature was maintained at 32 °C ± 0.5 °C. The diffusion medium samples were withdrawn at 2, 4, 6, 8, 12, 24, and 30 h and replaced with 0.5 mL of fresh diffusion medium. The samples were mixed with an internal standard solution, hexane, and vortexed for 1 min. The organic layer was separated using a dry-ice–acetone bath and then evaporated to dryness under airflow. Following reconstitution, the samples were transferred to a UPLC-UV vial for the analysis of the total absorption (sum of all diffusion medium samples) as well as the rate of absorption (transdermal flux). The ACQUITY UPLC^®^ system (Waters Corporation, Milford, MA, USA) included a UPLC column Acquity UPLC BEH C18 (100 mm × 2.1 mm, 1.7 µm).

The distribution of testosterone within the skin samples was determined at the end of the 30 h diffusion process. The skin samples were washed three times with an internal standard solution, and the resulting washed sample solutions were diluted with 50% ethanol to fit within the range of the calibration curve. The skin was then tape-stripped five times to remove the stratum corneum using 3M Transpore^®^ surgical tape. The tapes of each striped skin were collected into 20 mL of internal standard solution, vortexed for 30 min, and diluted 20 times. Following centrifugation, the resulting clear solutions were transferred to a UPLC-UV vial to analyze the skin content (stratum corneum). 

The stripped skin samples were divided into two layers: epidermis and dermis. Each layer was cut into small pieces and transferred to separate test tubes. To each tube was added 10 mL of the internal standard solution; then, the tubes were vortexed for 30 min and centrifuged at 14,000 rpm for 10 min. The resulting clear solutions were transferred to a UPLC-UV vial to analyze the skin content (epidermis and dermis).

### 2.3. Analytical Testing

The topical gels were formulated in Atrevis hydrogel, which is considered a preserved aqueous vehicle. The USP General Chapter <795> “Pharmaceutical Compounding—Nonsterile Preparations” defaults the BUD of preserved aqueous dosage forms to be 35 days in the absence of a USP-NF compounded preparation monograph, or compounded nonsterile preparation-specific stability information [12]. Data from a stability study using a stability-indicating analytical method would allow the BUD to be extended to the timeframe indicated by the study. To establish a longer BUD on the topical gel formulations, a stability study was conducted.

The sample preparation consisted of setting two batches of 300 g for testosterone 0.5% topical gel (lots 07272017-01 and 07272017-02) and two batches of 300 g for testosterone 20% topical gel (lots 07272017-03 and 07272017-04). The method of preparation is detailed in Table 1. Testosterone USP Micronized CIII (Soy) (lot C181188), Propylene Glycol USP (lot C179804), and the topical base Atrevis hydrogel (lot 20170518) were all obtained from PCCA (Houston, TX, USA). Following preparation, the topical gels were evenly distributed into ten 30 g plastic pumps. According to USP General Chapter <795> “Pharmaceutical Compounding—Nonsterile Preparations”, the suggested storage temperature for preserved aqueous dosage forms is controlled room temperature or refrigerator [12]. To assess the stability of testosterone in both storage conditions, one batch of each strength was stored in an environmentally controlled chamber (Thermo-Scientific, Waltham, MA, USA, model number 3940) at a relative humidity of 60% ± 5% and a temperature of 25 °C ± 2 °C (also referred to as controlled room temperature), and the other batch of each strength was stored at a temperature of 5 °C ± 3 °C in another environmentally controlled chamber (Thermo-Scientific, model number 3940). The temperature and humidity were monitored and registered daily.

For the physicochemical stability testing of the topical gels, one test pump was withdrawn from each storage condition (refrigerated temperature and controlled room temperature) at pre-determined time points, as follows: days 0, 7, 14, 28, 42, 60, 90, 123, and 182. Sampling was performed by pumping/dispensing directly from the pumping device.

Physical characteristics: The physical characteristics of the topical gels were evaluated by visually inspecting the samples for any changes in color and appearance, observing for odor change, and testing for pH and viscosity. The observations and values determined on day 0 were used as the basis for any significant change. To assess the appearance and odor change, each sample was evenly spread on a watch glass for visual inspection. To determine the color, each sample was compared against a Munsell color reference chart (Sigma-Aldrich, St. Louis, MO, USA) under an artificial daylight lamp. The pH measurements were made with a Horiba LaquaTwin pH meter (Kyoto, Japan), which was calibrated with certified pH 4.0 and 7.0 buffer solutions before each use. The viscosity of the topical gels was obtained using a RheoSense portable viscometer (microVISC, San Ramon, CA, USA). A microVISC pipette was used to draw about 1 mL of sample. The pipette was inserted into the viscometer, and the measurement was run on the automatic setting.Chemical characteristics: An UPLC method was validated to quantitate the testosterone in the topical gels. The assay determined on day 0 was used as the basis for any significant change. The stability-indicating UPLC method utilized a certified Waters Acquity H-class UPLC system (Milford, MA, USA) equipped with a separation module, a column heater/cooler, an auto-sampler, and an ultraviolet photo-diode array (PDA) detector. The chromatographic method employed a Waters Acquity CSH Phenyl-Hexyl C18 1.7 µm 2.1 mm × 100 mm column (part number 186005407; lots 01093516916627 and 01123628618226). The chromatographic data were acquired and processed using the Waters Empower 3 software.

The UPLC assay testing was designed by the ICH “Harmonised Triparty Guideline: Stability Testing of New Drug Substances and Products Q1A(R2)” [13]. It consists of a reverse phase, gradient chromatographic method with two different mobile phases (A:B) at a ratio of 70:30. The mobile phase A was 0.1% trifluoroacetic acid (lot 55267541; EMD Chemicals Inc., Gibbstown, NJ, USA) in water, whereas the mobile phase B was 0.1% trifluoroacetic acid in acetonitrile (part 34998-4L; Sigma-Aldrich, St. Louis, MO, USA). The method ran for 5.5 min with a flow rate of 0.5 mL/min. The column was heated to 50 °C, while the sample vials were stored at 6 °C in the autosampler. The standard solutions and sample solutions were injected into the separation module at a volume of 1 µL. Chromatographic data were acquired in 3-dimensions from 190 to 400 nanometers. The detection and quantitation of testosterone were performed at 245 nanometers. 

The samples for the UPLC were prepared by weighing approximately 0.5 g of each topical gel (testosterone 0.5% and 20%) into a 50 mL centrifuge tube and diluting the gels with 39.5 mL of methanol. Following vortex-mixing and sonication, the solution was centrifuged for 10 min at 6000 rpm. The supernatant was pipetted into a 2 mL tube and micro-centrifuged at 14,000 rpm for 10 min. Finally, the supernatant was transferred to a UPLC vial for assay testing.

3.Method validation: The UPLC assay testing method was validated by analyzing the following parameters: system suitability, linearity, accuracy, precision (repeatability and intermediate), robustness, solution stability, and specificity. The results obtained met the established acceptance criteria for all the parameters. As such, the UPLC assay testing method indicates stability for the compounded testosterone gels. A summary of the parameters, acceptance criteria, and results for validating the UPLC assay testing method is displayed in the Appendix A.

## 3. Results and Discussion

### 3.1. Case Study

The patient answered all questions of the ADAM questionnaire twice by referring to his late-onset hypogonadal symptoms before and after treatment with the compounded testosterone gel. Before treatment, a positive ADAM score was obtained since the patient answered ‘Yes’ to questions 1 and 7, which refer to the self-reported decreased libido and decreased strength of erections. According to this screening test, the patient suffered from male late-onset hypogonadism at baseline. After treatment, a negative ADAM score was obtained since the patient answered ‘No’ to all 10 questions. As such, it is assumed that the patient’s androgen deficiency improved with the testosterone therapy. 

Regarding treatment safety, the patient did not report any side effects. In addition, the patient’s PSA (prostate-specific antigen) did not increase while on testosterone supplementation. The patient reported performing the PSA test annually at the routine urologist visit since his sixties. Nevertheless, according to the latest guidelines, the American Urological Association (AUA) recommends against the routine PSA screening of men older than 70 years [14]. 

The laboratory result obtained after treatment for the patient’s serum total testosterone was 1076 ηg/dL. The reference range used by the testing laboratory for testosterone is 250–1100 ηg/dL. According to this range, the level achieved is clinically optimal. However, there is a significant variation in the reference ranges for total testosterone among testing laboratories [11]. There is also conflicting information regarding the ideal method to evaluate the patient’s hormone levels (serum, saliva, or urine testing) [15]. According to Livingston et al. [11], significant evidence supports that healthy reference ranges based on population distributions should not be used. Hormone levels are patient-specific, and the results obtained should be correlated with the clinical assessment. 

A limitation to consider in this case study is the single data point of serum total testosterone. Ideally, more data points should have been obtained, such as baseline and at 3-months post-treatment, but the retrospective methodology limited the study to the data that had been previously collected. 

In this case study, the patient and the physician stated that the results obtained with the testosterone therapy were beyond their expectations. 

### 3.2. Biological Testing

The rate of percutaneous absorption and the distribution of testosterone across the skin layers were determined by the UPLC-UV quantitative method. The rate of percutaneous absorption is presented as a flux, a time-averaged value determined across the sampling period and reported at the mid-point of sample collection (1, 3, 5, 7, 10, 18, and 27 h). Figure 1 illustrates the mean flux (ug/cm^2^/hour) plotted as the amount of testosterone absorbed through the skin over time. Upon dose application, it is observed that there is a rise in flux to a peak of testosterone at approximately 8–12 h across all samples, followed by a slow decline. The testosterone penetration profiles were essentially similar for the majority of samples, though differing in magnitude. Sample 2 showed the most significant penetration (11.53 µg/30 h) through the skin, whereas sample 9 showed the least penetration (4.03 µg/30 h).

The data obtained from the in vitro Franz finite dose model indicated that testosterone in all the ten cream formulations can penetrate into and through ex vivo human skin.

The distribution of testosterone following a dose exposure of 30 h to ex vivo human torso skin is presented as mass recovered per dose area. It is observed that the dose applied to the skin’s outer surface permeates into and through the stratum corneum, followed by the epidermis and then the dermis, to reach the receptor medium at last. As shown in Figure 2, most of the applied dose was retained in the skin’s stratum corneum, accounting for 64–88% across all formulations. The receptor medium content was in the range of 0.4–1.1%, whereas the dermal content was in the range of 0.5–1.6%, and the epidermis content in the range of 1.6–9.4%. The overall mass balance was outstanding, in the range of 99–104% of the applied dose across all ten formulations.

A limitation to consider in biological testing is the in vitro methodology of the study. The ideal setting to evaluate drug permeation is an in vivo human, hence the relevance of the case study described in this multidisciplinary study. Nevertheless, skin models are increasingly used for the testing of transdermal drugs, including those using ex vivo human skin [16]. These studies may only be considered a prediction of the in vivo skin permeation and, although a correlation exists, there are differences to consider when drugs permeate vascularized skin.

### 3.3. Analytical Testing

The physical characteristics of the testosterone gels were all within the specifications. The testosterone 0.5% topical gel exhibited a faint beige color, whereas the testosterone 20% topical gel exhibited a white color. Both gels presented a smooth, homogeneous appearance and a characteristic odor. A slight separation was noted on the top of the container for the testosterone 0.5% topical gel, stored at a refrigerated temperature from day 123 onwards, but it was considered insignificant. The ranges of viscosity and pH for the testosterone 0.5% topical gel were as follows: 1044.7–1146.0 mPa.s and 5.69–6.00 (room temperature) and 1025.2–1168.0 mPa.s and 5.54–5.89 (refrigerated conditions), respectively. The ranges of viscosity and pH for the testosterone 20% topical gel were as follows: 3452.0–6595.0 mPa.s and 5.67–6.05 (room temperature) and 3933.7–6595.0 mPa.s and 5.74–6.05 (refrigerated conditions), respectively.

The chemical characterization of the topical gels was conducted using UPLC assay testing, which measured the central chromatographic peak, as shown in Figure 3. It provided the mean strength of testosterone per time point. Both the testosterone 0.5% and 20% topical gels, stored at room temperature and refrigerated conditions, remained within the United States Pharmacopeia-National Formulary (USP-NF) specifications of ±10% variation in limits (90–110%) for the duration of the study [12]. The strength of the testosterone 0.5% gel, at room temperature and refrigerated conditions, varied in the range of 98.23–101.70% and 97.83–102.09%, respectively (Figure 4). Likewise, the strength of the testosterone 20% gel, at room temperature and refrigerated conditions, varied in the range of 98.57–102.61% and 98.90–102.00%, respectively (Figure 5).

As a result, the testosterone topical gels are physically and chemically stable at both storage conditions for 6 months. The physicochemical characterization of the testosterone 0.5% and 20% topical gels stored at room temperature and refrigerated conditions is reported in detail in the Appendix A.

## 4. Conclusions

Testosterone therapy may be a key treatment option in andropause patients with laboratory evidence for low testosterone and self-reported symptoms of androgen deficiency. Although not a life-threatening condition, androgen deficiency may have a considerable negative impact on the quality of life of men. Pharmacists are uniquely positioned to work with these patients and their physicians to achieve better treatment outcomes and overall health benefits by providing personalized topical compounded medications. As shown in this case report, a compounded testosterone gel of 1–5% is presented as a promising topical treatment option for testosterone therapy to benefit the complexity of low testosterone symptoms (androgen deficiency questionnaire) and increase the total serum testosterone level (laboratory results). The clinical efficacy is consistent with the results obtained in the biological testing. It is shown in vitro that testosterone is able to penetrate into and through human skin, which correlates with the results obtained in clinical practice. The analytical testing assures pharmacists of a quality formulation with an extended stability of 6 months. As such, compounded testosterone 0.5–20% topical gels are deemed to be stable and effective when treating late-onset hypogonadism.

## Figures and Tables

**Figure 1 pharmaceutics-16-00621-f001:**
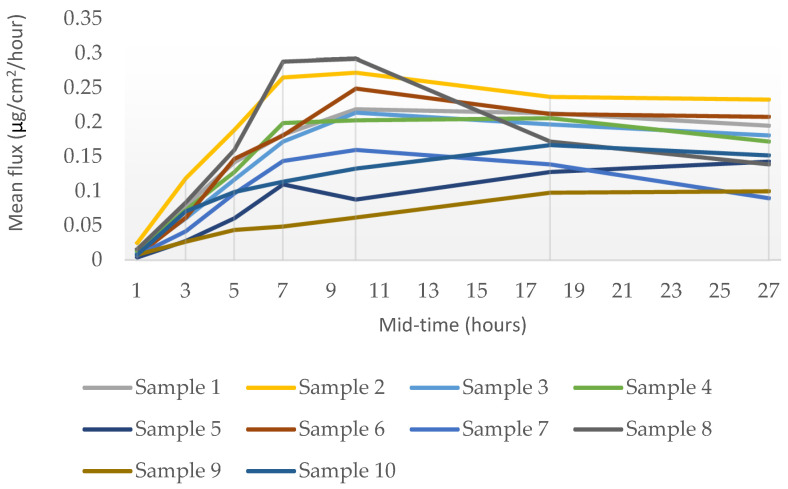
Rate of the percutaneous absorption (mean flux) of testosterone through ex vivo human skin over 30 h from a single application (mean, n = 3 donors).

**Figure 2 pharmaceutics-16-00621-f002:**
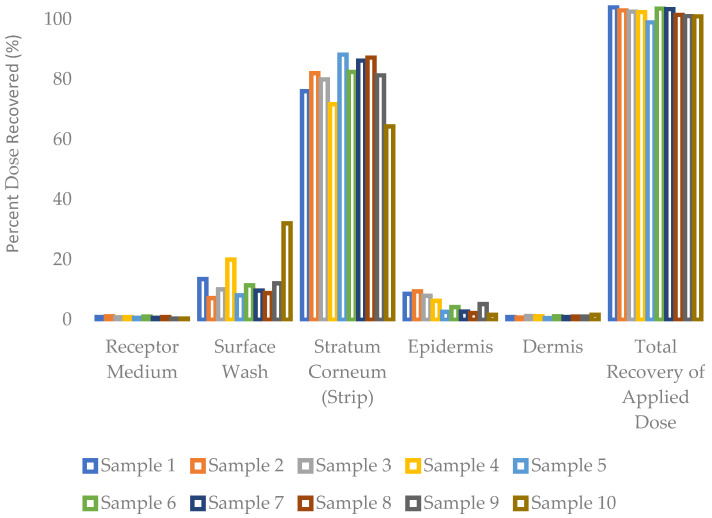
Distribution of testosterone into and through ex vivo human skin over 30 h from a single application (mean, n = 3 donors).

**Figure 3 pharmaceutics-16-00621-f003:**
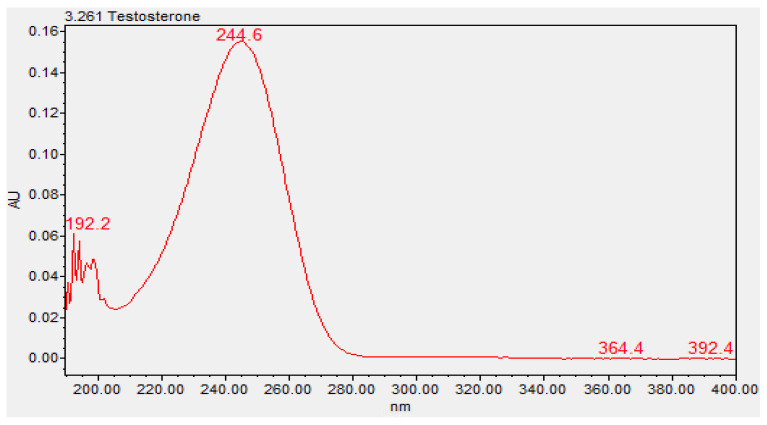
Ultraviolet spectrum of testosterone (3.261, Waters Acquity PDA Detector).

**Figure 4 pharmaceutics-16-00621-f004:**
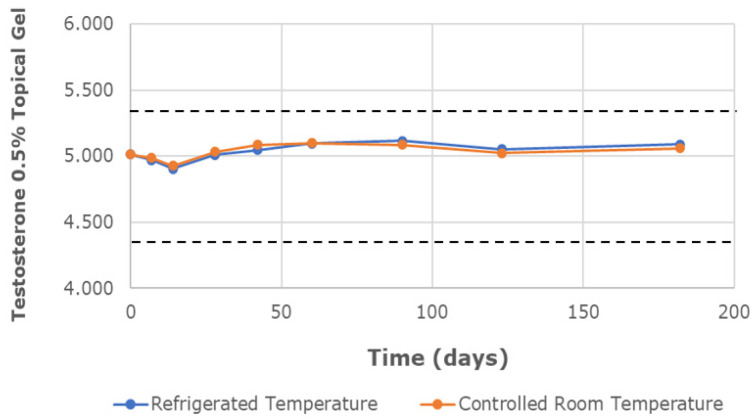
The mean testosterone concentration was 0.5% in the topical gel, stored at refrigerated and controlled room temperatures, over a study period of 6 months. Dashed lines represent the lower and upper limits, corresponding to 90% and 110% of the labeled concentration (5 mg/g), respectively.

**Figure 5 pharmaceutics-16-00621-f005:**
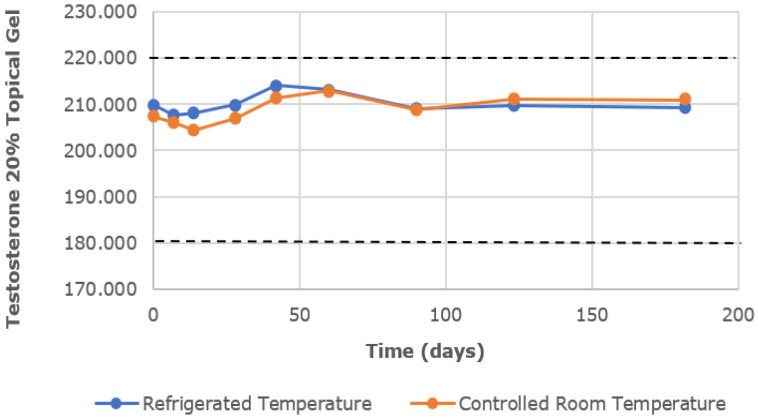
The mean testosterone concentration was 20% in the topical gel, stored at refrigerated and controlled room temperatures, over a study period of 6 months. Dashed lines represent the lower and upper limits, corresponding to 90% and 110% of the labeled concentration (200 mg/g), respectively.

**Table 1 pharmaceutics-16-00621-t001:** Formula, method of preparation, and storage instructions for a testosterone 5% topical gel (100 g).

Formula for 100 g	
Testosterone USP Micronized CIII (Soy)	5 g
Propylene Glycol USP	10 g
Base, Atrevis Hydrogel^®^	85 g
Method of Preparation	
Calculate the required quantity of each ingredient for the total amount to be prepared. Accurately weight each ingredient. Prepare a paste of testosterone in propylene glycol. Add the topical base and mix well until a homogeneous, slightly-off-white gel is obtained.	
Storage Instructions Store in an air-tight, light-resistant container. Protect from light. Store at a controlled room temperature of 20–25 °C.	

## Data Availability

The data can be shared up on request.

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
