# Peer review of "Testosterone Therapy for Late-Onset Hypogonadism: A Clinical, Biological, and Analytical Approach Using Compounded Testosterone 0.5–20% Topical Gels"

_pharmaceutics, 2024, doi:10.3390/pharmaceutics16050621_

Round 1
Reviewer 1 Report
Comments and Suggestions for Authors
Review of "Compounding for Testosterone Replacement Therapy in Male Hypogonadism: A Clinical, Biological, and Analytical Approach"
Review Assessment:
The article provides a comprehensive and well-structured exploration of compounded testosterone replacement therapy for male hypogonadism. Including a case study, biological testing, and analytical testing strengthens the validity of the findings. The methodology and results are presented. The data support the conclusions, and the study contributes valuable insights into the potential of compounded testosterone gels in clinical practice.
Summary:
The article explores the clinical, biological, and analytical aspects of compounded testosterone replacement therapy (TRT) for male hypogonadism. It includes a retrospective case study, biological testing on skin permeation, and analytical testing for stability. The study evaluates the efficacy of a compounded testosterone 5% topical gel in a hypogonadal patient in his seventies, considering both clinical symptoms and laboratory results.
Key Points:
1. Introduction:
· Male hypogonadism is characterized by low testosterone, leading to various symptoms affecting sexual and overall health.
· The study focuses on compounded testosterone gels (1-5%) as a potential alternative for TRT.
2. Case Study:
· The case study involves a male patient in his seventies using a compounded testosterone 5% topical gel.
· Clinical efficacy is assessed through the Androgen Deficiency in the Aging Male (ADAM) questionnaire and serum total testosterone testing.
· Results indicate improved ADAM scores and a clinically optimal serum total testosterone level.
3. Biological Testing:
· In vitro permeation testing (IVPT) is conducted on human cadaver skin to evaluate the skin permeation of testosterone 10% compounded topical gel.
· The study demonstrates the ability of testosterone to penetrate ex vivo human skin with a high degree of mass recovery.
4. Analytical Testing:
· Stability testing for compounded testosterone 0.5% and 20% topical gels is performed for six months under different storage conditions.
· Both physical and chemical stability are assessed, including color, odor, pH, and UPLC assay testing.
· Results indicate that both testosterone formulations remain within USP-NF specifications, ensuring stability over the study period.
Specific review.
Title:
The title communicates the focus of the manuscript. However, it could be more specific by including the concentration of the compounded testosterone gel, which is crucial information for readers interested in the study.
Abstract:
Positive Aspects:
- Provides a clear overview of the study's objectives, methods, and key findings.
- Clearly states the importance of testosterone in men's health and the relevance of the study in addressing hypogonadism.
- Keywords provide a concise summary of the main themes.
Areas for Improvement:
- The abstract could benefit from including more specific results or outcomes, offering readers a glimpse into the findings.
- The abstract is somewhat brief; expanding it slightly to incorporate more details on the significance of the study might enhance its impact. The journal's rules for the abstract are 200 words, and the manuscript abstract is 187.
Introduction:
Positive Aspects:
- Clearly defines male hypogonadism and its impact on health.
- Establishes the rationale for exploring compounded testosterone gels, providing a context for the study.
- Sets the stage for the multidisciplinary approach taken in the study.
Areas for Improvement:
- While the introduction effectively outlines the problem, it could be strengthened by presenting more recent statistics or studies on the prevalence and impact of hypogonadism.
- A more explicit statement about the research gap or the specific contribution this study makes to existing literature would enhance the introduction.
Case Study:
Positive Aspects:
- The detailed case study provides essential information about the patient, treatment, and assessment methods.
- Including the ADAM questionnaire and laboratory testing adds depth to the clinical evaluation.
Areas for Improvement:
- The case study's limitations and potential biases should be acknowledged to enhance transparency.
- The rationale for choosing a retrospective approach and any potential impact on the study's validity could be discussed.
Biological Testing:
Positive Aspects:
- The in vitro permeation testing is well-described, providing clarity on the methodology.
- The use of human cadaver skin adds relevance to the biological testing.
Areas for Improvement:
- More discussion on the clinical implications of the biological testing results and their alignment with the case study findings could enhance this section.
Analytical Testing:
Positive Aspects:
- Comprehensive details on the methodology of stability testing are provided.
- Inclusion of physical and chemical stability parameters adds robustness to the analytical testing.
Areas for Improvement:
- A more extensive discussion on the practical implications of the stability results would enhance the value of this section.
- Clarification on why specific storage conditions were chosen would provide insight into the practical application of the results.
Results and Discussion:
Positive Aspects:
- Presents a clear summary of the case study and testing outcomes.
- Connects the clinical, biological, and analytical findings, reinforcing the study's multidisciplinary approach.
- Conclusions are well-supported by the results.
Areas for Improvement:
- A more detailed exploration of potential confounding factors or limitations in the study design would strengthen the discussion.
- Comparisons with existing literature or studies could enhance the contextualization of the study's findings.
Supplementary Materials:
Positive Aspects:
- Including supplementary materials provides additional data and details for interested readers.
- Tables and figures in the supplementary materials are clear and well-labeled.
Areas for Improvement:
- A brief explanation or summary at the beginning of the supplementary materials could help guide readers through the content more effectively.
Overall Assessment:
The manuscript is well-structured and comprehensively covers essential aspects of compounded testosterone replacement therapy. Integrating clinical, biological, and analytical perspectives adds depth to the study. Further elaboration on certain points, including limitations and practical implications, would enhance the manuscript's overall impact and appeal to a broader audience. The clarity of language and organization make the content accessible, and the study contributes valuable insights to the field of testosterone replacement therapy for male hypogonadism.
Conclusions:
The study concludes that compounded testosterone gel (1-5%) is a promising option for TRT, demonstrating clinical efficacy, skin permeation, and stability. The compounded formulations show extended stability, supporting their use beyond the usual beyond-use date. The article emphasizes the potential of personalized, topical TRT in managing hypogonadal symptoms.

I enclose the manuscript with suggestion.
Author Response
Please see the attachment, thank you.

Reviewer 2 Report
Comments and Suggestions for Authors
My major concerns are as follows:
1. The contents could not support the title. The research was focused only on hypogonadal patient in his seventies (line 12) or over 40 years of age (line 75). Therefore, “middle-aged or aging” might be important word and should be appeared in title. In addition to that, “patient in his seventies (line 12) or over 40 years of age” was also needed to clarify.
2. Abstract was not written in good forms and should be re-organized. The key problem for abstract was too general and lake of scientific data and key results.
3. It was not wise for the authors used only ADAM questionnaire (too simple and suitable for primary screening) to evaluate patients’ testosterone deficiency symptoms, especially the research needed to evaluate the effectiveness of testosterone supplement therapy (TST). Therefore, AMS questionnaire might be more suitable. Why did the authors not use AMS questionnaire?
4. It was better to use testosterone supplement therapy (TST) rather than TRT, because middle-aged male hypogonadism patients had some testosterone in his body, and replace (R) was not accurate.
5. Generally accepted standard for normal testosterone level was over 300 ng/dL. It was usual to increase the testosterone level to 1,076 ng/dL (line 210). This might suggest that the over large dosage for TST (it was no need and with high side effective possibility). In addition to that, the testing point (to test testosterone) was also very important. The research might be done within the treatment period. I also concern that if the TST is finished for a period (for instance 3 months), how about the testosterone level for these patients, might be even lower?
6. The most important problem was safety. With large dosage and long period TST (6 months, line 86), did the authors concern prostate safety including PSA and other safety aspects?
7. References were not enough, and many related key references should be supplemented.
Comments on the Quality of English Languagelanguage was acceptable, and simple editing was enough.
Author Response
Please see responses attached, thank you.

Reviewer 3 Report
Comments and Suggestions for Authors
On lines 33,34, the statement, "It has been suggested that it is also associated with prostate cancer and benign prostatic hyperplasia [2]" is not supported by the reference, nor by guideline summaries of data and should be deleted.
Worth adding oral preparations in introduction.
It is not clear how the case study is relevant to this in vitro evaluation of penetration of cadaveric skin.
It is not clear how well the in vitro study reflects absorption from vascularized skin, a limitation to emphasize in the discussion.
The purpose of the study should be more clearly elucidated early in the manuscript.
It is not clear how a single serum measurement of testosterone after 6 months of treatment is useful as a measure of testosterone absorption.
Again, the results of a single time point evaluation in the case study appear incongruent and irrelevant to the in vitro study being done. THe authors could have made much more of that evaluation by looking at different doses, different times after administration, etc. Although dose adjustments are routinely done during testosterone replacement therapy, a serum level of 1,076 ng/dL is considered higher than the target range of 400-600 in most guidelines.
The stability data were quite interesting.
The authors should discuss whether testosterone is absorbed from stratum corneum, since that's where the topical testosterone was localized.
Comments on the Quality of English Language
Some improvement is possible (mis-spellings, clearer statements and organization of the manuscript.
Author Response
Please see the attachment, thank you.

Round 2
Reviewer 2 Report
Comments and Suggestions for Authors
1. I feel so sorry for a new question that I neglected last time. The concept in title (ageing male hypogonadism) is not proper. there is already a standard name for it that is LOH (late onset hypogonadism), and the concept is well accepted in scientific area. Therefore, some contents should be changed accordingly, such as contents in line 28, from “hypogonadal patient” to “a LOH patient”.
2. Considering the other questions, proper explanation or reply has been given or as limitation (AMS questionnaire and PSA testing results cannot be supplemented before and after TST) presented in discussion part, and references supplemented.
3. Considering the side effect, only with explanation “Regarding treatment safety, the patient did not report any side effects” (contents in line 279) is not enough. Both physicians and LOH patients with TST are more care about prostate safety and prostate cancer, and monitoring PSA (prostate specific antigen) before and after TST should be mandatory. The author should explain the limitation in discussion part.
Author Response
Please see attached, thank you.
